# Newborn Screening Today and Tomorrow: A Brief Report from the International Primary Immunodeficiencies Congress

**DOI:** 10.3390/ijns10020030

**Published:** 2024-04-05

**Authors:** Leire Solis, Samya Van Coillie, James R. Bonham, Fabian Hauck, Lennart Hammarström, Frank J. T. Staal, Bruce Lim, Martine Pergent, Johan Prévot

**Affiliations:** 1IPOPI, BE-1050 Brussels, Belgium; leire@ipopi.org (L.S.); samya@ipopi.org (S.V.C.); bruce@ipopi.org (B.L.); martine@ipopi.org (M.P.); 2International Society of Neonatal Screening, Reigerskamp 273, 3607 HP Maarssen, The Netherlands; j.bonham@nhs.net; 3Division of Pediatric Immunology and Rheumatology, Department of Pediatrics, Dr. von Hauner Children’s Hospital, University Hospital, Ludwig-Maximilians-Universität München, DE-80337 Munich, Germany; fabian.hauck@med.uni-muenchen.de; 4Department of Medical Biophysics and Biochemistry, Karolinska Instituet, SE-17177 Stockholm, Sweden; lennart.hammarstrom@ki.se; 5Departments of Immunology and Pediatrics, Leiden University Medical School, 2300 RC Leiden, The Netherlands; f.j.t.staal@lumc.nl

**Keywords:** newborn screening, population screening, severe combined immunodeficiency (SCID), rare disorders

## Abstract

This article presents the report of the session on “Newborn Screening for Primary Immunodeficiencies—Now What?” organised during the International Primary Immunodeficiency Congress (IPIC) held in November 2023. This clinical conference was organised by the International Patient Organisation for Primary Immunodeficiencies (IPOPI), the global patient organisation advocating for primary immunodeficiencies (PIDs) in patients. The session aimed at exploring the advances in newborn screening (NBS) for severe combined immunodeficiency, starting with the common practice and inserting the discussion into the wider perspective of genomics whilst taking into consideration the ethical aspects of screening as well as incorporating families and the public into the discussions, so as to ensure that NBS for treatable rare disorders continues to be one of the major public health advances of the 20th century.

We write to share key insights from the recent International Patient Organisation for Primary Immunodeficiencies (IPOPI) clinical conference, in light of the active role that IPOPI plays in global newborn screening (NBS) advocacy. As a proud member of the Screen4Rare coalition, alongside the International Society for Neonatal Screening (ISNS) and the European Society for Immunodeficiencies (ESID), IPOPI is committed to advancing NBS practices worldwide, emphasising a focus on well-defined and treatable conditions for which early detection and treatment during childhood significantly improve outcomes [1]. 

At the International Primary Immunodeficiencies Congress (IPIC) held in November 2023, experts convened for a session titled “Newborn Screening for Primary Immunodeficiencies—Now What?” to delve into critical aspects of NBS, particularly in the context of severe combined immunodeficiency (SCID). This session facilitated discussions on the current state of NBS, offering valuable insights into the challenges and opportunities in the field. The session was structured around the interventions of three experts in the field: Prof. James Bonham on “Newborn screening as a system: from good practice to common practice”, Prof. Fabian Hauck on “SCID newborn screening—What have we learnt and where do we go from here?” and Prof. Lennart Hammarström on “PIDs & NBS in 2030 (from new candidate diseases to the use of genomics)”. The session included the interventions of Mr. Guillaume Cordova and Ms. Solene Blouin, parents of two babies affected by SCID in France, and was moderated by Prof. Frank Staal and Mr. Bruce Lim. 

Since Dr. Guthrie’s pioneering work on phenylketonuria (PKU) and the consequent introduction of a screening programme in the US in the 1960s, 745 million newborns have been screened for PKU, allowing the identification and treatment of 62,000 infants with life-changing interventions [2]. However, the global NBS landscape exhibits significant variations, with differences in the number of diseases screened, programme coverage, and implementation effectiveness. For instance, SCID screening practices range from national implementation in the United States, Germany, Israel, New Zealand, The Netherlands, Sweden, Ireland, Denmark, Finland, Norway, Iceland, Czech Republic, Switzerland [3], Canada, and Japan to regional/provincial implementation in Italy, Spain, Vietnam, and Australia, with ongoing pilot projects in France and the United Kingdom [4]. Yet, the majority of countries worldwide still do not screen for SCID.

The fundamental aim of NBS is to ensure swift diagnosis, treatment, and care for infants affected by specific conditions. However, the expansion of NBS programmes to incorporate new diseases demands careful assessment and planning. Considering the Wilson and Jungner principles, which have formed the primary road map towards designing a screening policy since the late 1960s [5], the session stressed the importance of evidence-based decision-making, careful implementation, and collaboration among stakeholders, including screening and medical experts, patient representatives, and decision-makers, as exemplified by Prof. Fabian Hauck in his description of how SCID NBS was planned and implemented in Germany.

This is of particular importance in the context of the upcoming genomics evolution and the challenges and opportunities it presents for NBS, as was highlighted by Prof. James Bonham in his intervention. As technology advances, next-generation sequencing is starting to be applied in NBS, thereby significantly increasing the number of disorders that can be detected, either in a targeted or genome-/exome-wide manner. Future NBS programmes may even be based on whole genome sequencing (WGS) approaches able to detect a maximal number of disease-associated variants applied to unselected—assumed to be healthy—newborns [6,7,8]. While this would be a powerful diagnostic tool, it would not only bring about challenges regarding technical feasibility, interpretation of results, and economic cost, but it would also raise important ethical, legal, and social implications requiring careful consideration [9,10]. 

Despite the many challenges posed by next-generation sequencing and genomics evolution, they have already permeated many NBS and pilot programmes or will so in the coming future, as illustrated by Prof. Hammarström in his presentation, showing the studies conducted on WGS on unselected newborns in different countries. It is therefore necessary for all those stakeholders involved (professional societies, relevant patient organisations, health policy makers, and politicians) to start discussing the development of appropriate, well-organised, and equitable NBS programmes offered on a voluntary and informed basis to families to help identify well-defined treatable conditions where it is clear that their early asymptomatic detection and treatment during childhood results in significantly improved outcomes.

Recent case examples of the inclusion of SCID in NBS programmes worldwide, discussed during the session, underscored the importance of healthcare system preparedness for the inclusion of a new disease and the need for timely evaluation to ensure that the programme brings more benefit than harm [11]. For SCID, it has been shown that earlier diagnosis and consequently earlier therapeutic intervention through allogenic stem cell transplantation or autologous gene therapy have significantly improved the overall survival of treated SCID babies [12,13]. Indeed, the importance of working towards equity and innovation in NBS has been highlighted by several initiatives, including the Screen4Rare coalition and the recently launched European Reference Network (ERN) Expert Platform for Newborn Screening, and was a key action point identified during IPOPI’s 2022 Global Multi-Stakeholder’s Summit [14]. The Screen4Rare coalition further aims to promote and disseminate examples of good practice to support the organisation and conduct of NBS within the European Union [15].

In conclusion, this session from our recent clinical conference is part of IPOPI’s ongoing efforts to promote robust NBS practices, recognising the pivotal role it plays in identifying and treating rare diseases in newborns. We hope this conference report sparks further discussion and collaboration to enhance NBS practices globally.

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
