# Peer review of "Newborn Screening Today and Tomorrow: A Brief Report from the International Primary Immunodeficiencies Congress"

_2409-515X, 2024, doi:10.3390/ijns10020030_

Round 1

Reviewer 1 Report

Comments and Suggestions for Authors

This is a nice summary of newborn screening with a focus on immunodeficiencies and genetic screening.  While the content is not new, it is presented in a clear and concise manner.

Author Response

Thank you very much for taking the time to review our manuscript and provide your feedback. 

Reviewer 2 Report

Comments and Suggestions for Authors

There seems to be two different discussions in this letter: a discussion of SCID in NBS and a discussion of NGS in NBS. These concepts are poorly connected and both discussions merely state 'how things are' with few unique insights. A discussion of specific challenges that still face SCID internationally in light of rapidly advancing molecular competence in most NBS labs would be of benefit. If there are thoughts on how NGS may relate to SCID, this would also be of interest. The discussion of challenges regarding NGS in the NBS space have been noted in other spaces. 

Author Response

Thank you very much for taking the time to review and provide us with your comments and feedback. We have carefully assessed your comments and tried to address them. In that sense, we have tried to clarify what have the different speakers said during their intervention, so as to provide a clearer idea of what was covered during the session at IPIC. We have also tried to address the comment on why we speak about SCID NBS but also about NGS. We hope that we these changes, the text will be clearer. 
Thank you again for your review. 

Reviewer 3 Report

Comments and Suggestions for Authors

Dear authors,

thank you for your manuscript/report on the IPOPI congress. These congress reports are always very interesting and it can be accepted for publication, however, a few things should be considered:

1. Your manuscript does not fit into the category "Letter to the ditor", but much better into the category "Confernce reports".

You should take a look at the web site (https://www.mdpi.com/search?journal=IJNS&article_type=conference-report) and should restructure your manuscript a bit. Probably rewrite the first paragraph so that it will serve as an abstract.

2. In lines 46 to 50 you give a list of countries were SCID screening is already implemented. However, there is Austria (https://kinder-jugendheilkunde.meduniwien.ac.at/ueber-uns/neugeborenen-screening/)

and Switzerland (Swiss Med Weekly 2020 Jun 24:150:w20254. doi: 10. 4414/smw.2020.20254.eCollection 2020 Jun15) that have SCID screening as well, since 2019 and 2021 respectively. Switzerland is also listed on Ref. 3

You should check this carefully, in order that all countries get listed.

In addition, if the session on PID had several invitated speakers, it would be great if you could convince them to privide an abstract of their talk. These could be incorporated into your mansucript. 

Author Response

Thank you very much for taking the time to read our manuscript and to provide us with such clear feedback.
We have worked to address the comments and suggestions made as follow: 1) we have changed the format of the manuscript, so that it appears as "Conference report". We have reviewed the first paragraph to make of it an abstract and highlight the change of article type. 

2) Inclusion of Switzerland and other countries in the list of countries screening newborns for SCID at national level. However, in the link provided by this reviewer on NBS for Austria, we could not see SCID listed in the panel of diseases and could not find any other supportive material to also list this country. If any additional link is available and known by the reviewer, we would be very happy to include it in the manuscript. 

3) Abstracts from the speakers of the session: Some of the speakers in this session mentioned, when we started discussing the idea of having a manuscript about the session, that they would not like to see specific data to be included as their slides contained some materials that had been or would be used for publication. We have, however, specified the title of their talk and included some references to the ideas that they shared during the session, hoping that it would improve the overall understanding of the session. 

Round 2

Reviewer 2 Report

Comments and Suggestions for Authors

This is improved. The outcomes of the conference are clearer and its relationship to the current NBS landscape is better defined.